# Multistep Consistency Models

## Abstract

Diffusion models are relatively easy to train but require many steps to generate samples. Consistency models are far more difficult to train, but generate samples in a single step.

In this paper we propose Multistep Consistency Models: A unification between Consistency Models (Song et al., 2023) and TRACT (Berthelot et al., 2023) that can interpolate between a consistency model and a diffusion model: a trade-off between sampling speed and sampling quality. Specifically, a 1-step consistency model is a conventional consistency model whereas a $\infty$-step consistency model is a diffusion model.

Multistep Consistency Models work really well in practice. By increasing the sample budget from a single step to 2-8 steps, we can train models more easily that generate higher quality samples, while retaining much of the sampling speed benefits. Notable results are 1.4 FID on Imagenet 64 in 8 sampling steps and 2.1 FID on Imagenet128 in 8 sampling steps with consistency distillation, using simple losses without adversarial training. We also show that our method scales to a text-to-image diffusion model, generating samples that are close to the quality of the original model.

## 1 Introduction

Diffusion models have rapidly become one of the dominant generative models for image, video and audio generation (Ho et al., 2020; Kong et al., 2021; Saharia et al., 2022). The biggest downside to diffusion models is their relatively expensive sampling procedure: whereas training uses a single function evaluation per datapoint, it requires many (sometimes hundreds) of evaluations to generate a sample.

Recently, Consistency Models (Song et al., 2023) have reduced sampling time significantly, but at the expense of image quality. Consistency models come in two variants: Consistency Training (CT) and Consistency Distillation (CD) and both have considerably improved performance compared to earlier works. TRACT (Berthelot et al., 2023) focuses solely on distillation with an approach similar to consistency distillation, and shows that dividing the diffusion trajectory in stages can improve performance. Despite their successes, neither of these works attain performance close to a standard diffusion baseline.

Here, we propose a unification of Consistency Models and TRACT, that closes the performance gap between standard diffusion performance and low-step variants. We relax the single-step constraint from consistency models to allow ourselves as much as 4, 8 or 16 function evaluations for certain settings. Further, we generalize TRACT to consistency training and adapt step schedule annealing and synchronized dropout from consistency modelling. We also show that as steps increase, Multistep CT becomes a diffusion model. We introduce a unifying training algorithm to train what we call Multistep Consistency Models, which splits the diffusion process from data to noise into predefined segments. For each segment a separate consistency model is trained, while sharing the same parameters. For both CT and CD, this turns out to be easier to model and leads to significantly improved performance with fewer steps. Surprisingly, we can perfectly match baseline diffusion model performance with only eight steps, on both Imagenet64 and Imagenet128.

Another important contribution of this paper that makes the previous result possible, is a *deterministic* sampler for diffusion models that can obtain competitive performance on more

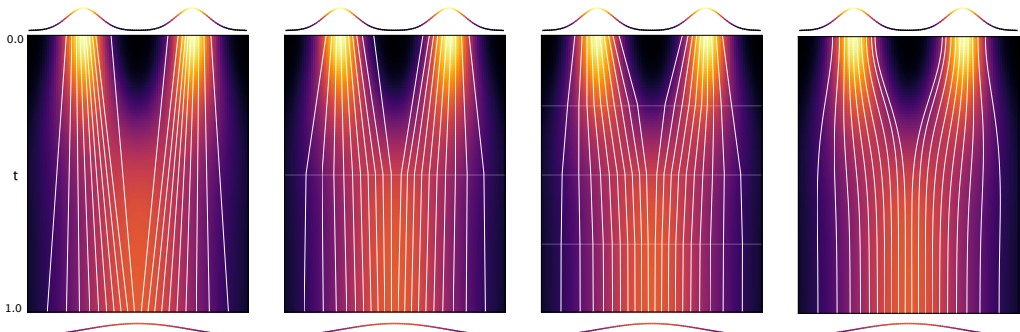

Figure 1: This figure shows that Multistep Consistency Models interpolate between (single step) Consistency Models and standard diffusion. Top at $t = 0$: the data distribution which is a mixture of two normal distributions. Bottom at $t = 1$: standard normal distribution. Left to right: the sampling trajectories of $(1, 2, 4, \infty)$-step Consistency Models (the latter is in fact a standard diffusion with DDIM) are shown. The visualized trajectories are real from trained Multistep Consistency Models. The 4-step path has a smoother path and will likely be easier to learn than the 1-step path.

complicated datasets such as ImageNet128 in terms of FID score. We name this sampler Adjusted DDIM (aDDIM), which essentially inflates the noise prediction to correct for the integration error that produces blurrier samples.

In terms of numbers, we achieve performance rivalling standard diffusion approaches with as little as 8 and sometimes 4 sampling steps. These impressive results are both for consistency training and distillation. A remarkable result is that with only 4 sampling steps, multistep consistency models obtain performances of 1.6 FID on ImageNet64 and 2.3 FID on Imagenet128.

## 2 BACKGROUND: DIFFUSION MODELS

Diffusion models are specified by a destruction process that adds noise to destroy data: $\boldsymbol{z}_t = \alpha_t \boldsymbol{x} + \sigma_t \boldsymbol{\epsilon}_t$ where $\boldsymbol{\epsilon}_t \sim \mathcal{N}(0,1)$. Typically for $t \to 1$, $\boldsymbol{z}_t$ is approximately distributed as a standard normal and for $t \to 0$ it is approximately $\boldsymbol{x}$. In terms of distributions one can write the diffusion process as: $q(\boldsymbol{z}_t|\boldsymbol{x}) = \mathcal{N}(\boldsymbol{z}_t|\alpha_t\boldsymbol{x}, \sigma_t)$.

Following (Sohl-Dickstein et al., 2015; Ho et al., 2020) we will let $\sigma_t^2 = 1 - \alpha_t^2$ (variance preserving). As shown in Kingma et al. (2021), the specific values of $\sigma_t$ and $\alpha_t$ do not really matter. Whether the process is variance preserving or exploding or something else, they can always be re-parameterized into the other form. Instead, it is their ratio that matters and thus it can be helpful to define the signal-to-noise ratio, i.e. $\text{SNR}(t) = \alpha_t^2/\sigma_t^2$. To sample from these models, one uses the denoising equation:

$$q(\boldsymbol{z}_s|\boldsymbol{z}_t, \boldsymbol{x}) = \mathcal{N}(\boldsymbol{z}_s|\mu_{t \to s}(\boldsymbol{z}_t, \boldsymbol{x}), \sigma_{t \to s}) \tag{1}$$

where $\boldsymbol{x}$ is approximated via a learned function that predicts $\hat{\boldsymbol{x}} = f(\boldsymbol{z}_t, t)$. Note here that $\sigma_{t \to s}^2 = \left(\frac{1}{\sigma_s^2} + \frac{\alpha_{t|s}^2}{\sigma_{t|s}^2}\right)^{-1}$ and $\boldsymbol{\mu}_{t \to s} = \sigma_{t \to s}^2\left(\frac{\alpha_{t|s}}{\sigma_{t|s}^2}\boldsymbol{z}_t + \frac{\alpha_s}{\sigma_s^2}\boldsymbol{x}\right)$ as given by (Kingma et al., 2021). In (Song et al., 2021b) it was shown that the optimal solution under a diffusion objective is to learn $\mathbb{E}[\boldsymbol{x}|\boldsymbol{z}_t]$, i.e. the expectation over all data given the noisy observation $\boldsymbol{z}_t$. One than iteratively samples for $t = 1, 1 - 1/N, \ldots, 1/N$ and $s = t - 1/N$ starting from $\boldsymbol{z}_1 \sim \mathcal{N}(0,1)$. Although the amount of steps required for sampling depends on the data distribution, empirically generative processes for problems such as image generation use hundreds of iterations making diffusion models one of the most resource consuming models to use (Luccioni et al., 2023).

**Consistency Models** In contrast, consistency models (Song et al., 2023; Song & Dhariwal, 2023) aim to learn a direct mapping from noise to data. Consistency models are constrained

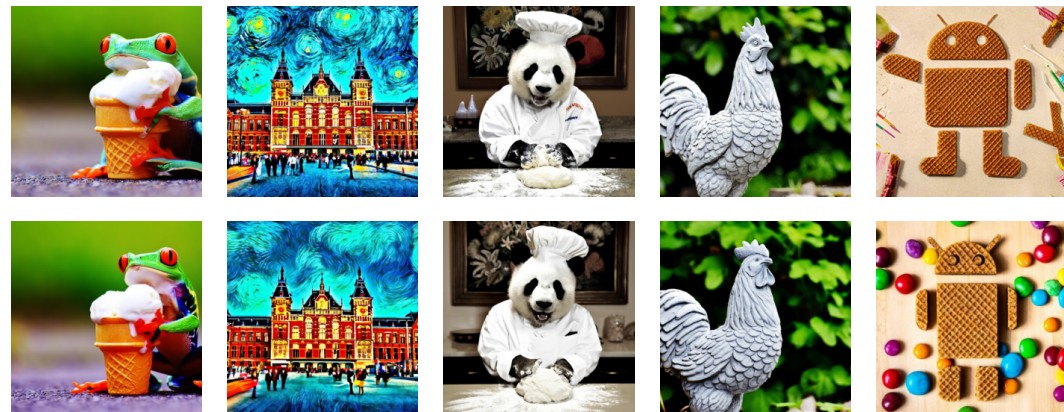

Figure 2: Qualititative comparison between a multistep consistency and diffusion model. Top: ours, samples from aDDIM distilled 16-step concistency model (3.2 secs). Bottom: generated samples usign a 100-step DDIM diffusion model (39 secs). Both models use the same initial noise.

to predict $\boldsymbol{x} = f(\boldsymbol{z}_0, 0)$, and are further trained by learning to be *consistent*, minimizing:

$$||f(\boldsymbol{z}_t, t) - \text{nograd}(f(\boldsymbol{z}_s, s))||, \qquad (2)$$

where $\boldsymbol{z}_s = \alpha_s \boldsymbol{x} + \sigma_s \boldsymbol{\epsilon}$ and $\boldsymbol{z}_t = \alpha_t \boldsymbol{x} + \sigma_t \boldsymbol{\epsilon}$, (note both use the same $\boldsymbol{\epsilon}$) and $s$ is closer to the data meaning $s < t$. When (or if) a consistency model succeeds, the trained model solves for the probability ODE path along time. When successful, the resulting model predicts the same $\boldsymbol{x}$ along the entire trajectory. At initialization it will be easiest for the model to learn $f$ near zero, because $f$ is defined as an identity function at $t = 0$. Throughout training, the model will propagate the end-point of the trajectory further and further to $t = 1$. In our own experience, training consistency models is much more difficult than diffusion models.

**Consistency Training and Distillation**  Consistency Models come in two flavours: Consistency Training (CT) and Consistency Distillation (CD). In the paragraph before, $\boldsymbol{z}_s$ was given by the data which would be the case for CT. Alternatively, one might use a pretrained diffusion model to take a probability flow ODE step (for instance with DDIM). Calling this pretrained model the teacher, the objective for CD can be described by:

$$||f(\boldsymbol{z}_t, t) - \text{nograd}(f(\text{DDIM}_{t \to s}(\boldsymbol{x}_{\text{teacher}}, \boldsymbol{z}_t), s))||, \qquad (3)$$

where DDIM now defines $\boldsymbol{z}_s$ given the current $\boldsymbol{z}_t$ and (possibly an estimate of) $\boldsymbol{x}$.

An important hyperparameter in consistency models is the gap between the model evaluations at $t$ and $s$. For CT large gaps result in a bias, but the solutions are propagated through diffusion time more quickly. On the other hand, when $s \to t$ the bias tends to zero but it takes much longer to propagate information through diffusion time. In practice a step schedule $N(\cdot)$ is used to anneal the step size $t - s = 1/N(\cdot)$ over the course of training.

**DDIM Sampler**  The DDIM sampler is a linearization of the probability flow ODE that is often used in diffusion models. In a variance preserving setting, it is given by:

$$\boldsymbol{z}_s = \text{DDIM}_{t \to s}(\boldsymbol{x}, \boldsymbol{z}_t) = \alpha_s \boldsymbol{x} + (\sigma_s / \sigma_t)(\boldsymbol{z}_t - \alpha_t \boldsymbol{x}) \qquad (4)$$

In addition to being a sampling method, the DDIM equation will also prove to be a useful tool to construct an algorithm for our multistep diffusion models.

Another helpful equations is the inverse of DDIM (Salimans & Ho, 2022), originally proposed to find a natural way parameterize a student diffusion model when a teacher defines the sampling procedure in terms of $\boldsymbol{z}_t$ to $\boldsymbol{z}_s$. The equation takes in $\boldsymbol{z}_t$ and $\boldsymbol{z}_s$, and produces $\boldsymbol{x}$ for which $\text{DDIM}_{t \to s}(\boldsymbol{x}, \boldsymbol{z}_t) = \boldsymbol{z}_s$. It can be derived by rearranging terms from the DDIM equation:

$$\boldsymbol{x} = \text{invDDIM}_{t \to s}(\boldsymbol{z}_s, \boldsymbol{z}_t) = \frac{\boldsymbol{z}_s - \frac{\sigma_s}{\sigma_t}\boldsymbol{z}_t}{\alpha_s - \alpha_t \frac{\sigma_s}{\sigma_t}}. \qquad (5)$$

## 3 Multistep Consistency Models

In this section we describe multi-step consistency models. First we explain the main algorithm, for both consistency training and distillation. Furthermore, we show that multi-step consistency converges to a standard diffusion training in the limit. Finally, we develop a deterministic sampler named aDDIM that corrects for the missing variance problem in DDIM.

### 3.1 General description

Multistep consistency splits up diffusion time into equal segments to simplify the modelling task. Recall that a consistency model must learn to integrate the full ODE integral. This mapping can become very sharp and difficult to learn when it jumps between modes of the target distribution as can be seen in Figure 1. A consistency loss can be seen as an objective that aims to approximate a path integral by minimizing pairwise discrepancies. Multistep consistency generalizes this approach by breaking up the integral into multiple segments. Originally, consistency runs until time-step 0, evaluated at some time $t > 0$. A consistency model should now learn to integrate the DDIM path until 0 and predict the corresponding $\boldsymbol{x}$. Instead, we can generalize the consistency loss to targets $\boldsymbol{z}_{t_{\text{step}}}$ instead

---

**Algorithm 1** Training Multistep CMs

Sample $\boldsymbol{x} \sim p_{\text{data}}$, $\boldsymbol{\epsilon} \sim \mathcal{N}(0, \mathbf{I})$, train iteration $i$
$N_{\text{per segment}} = \text{round}(N_{\text{teacher}}(i)/\text{student\_steps})$
$\text{step} \sim \mathcal{U}(0, \text{student\_steps} - 1)$
$n_{rel} \sim \mathcal{U}(1, N_{\text{per segment}})$
$t_{\text{step}} = \text{step}/\text{student\_steps}$
$\boldsymbol{x}_{\text{teacher}} = \begin{cases} \boldsymbol{x} & \text{if training} \\ f_{\text{teacher}}(\boldsymbol{z}_t, t) & \text{if distillation} \end{cases}$
$x_{\text{var}} = ||\boldsymbol{x}_{\text{teacher}} - \boldsymbol{x}||^2/d$
$t = t_{\text{step}} + n_{rel}/T$ and $s = t - 1/T$
$\boldsymbol{z}_t = \alpha_t \boldsymbol{x} + \sigma_t \boldsymbol{\epsilon}$
$\boldsymbol{z}_s = \text{aDDIM}_{t \to s}(\boldsymbol{x}_{\text{teacher}}, \boldsymbol{z}_t, x_{\text{var}})$
$\hat{\boldsymbol{x}}_{\text{ref}} = \text{nograd}(f(\boldsymbol{z}_s, s))$
$\hat{\boldsymbol{x}} = f(\boldsymbol{z}_t, t)$
$\hat{\boldsymbol{z}}_{\text{ref}, t_{\text{step}}} = \text{DDIM}_{s \to t_{\text{step}}}(\hat{\boldsymbol{x}}_{\text{ref}}, \boldsymbol{z}_s)$
$\hat{\boldsymbol{x}}_{\text{diff}} = \text{invDDIM}_{t \to t_{\text{step}}}(\hat{\boldsymbol{z}}_{\text{ref}, t_{\text{step}}}, \boldsymbol{z}_t) - \hat{\boldsymbol{x}}$
$L_t = w_t \cdot ||\hat{\boldsymbol{x}}_{\text{diff}}||$ for instance $w_t = \text{SNR}(t) + 1$

---

of $\boldsymbol{x}$ ($\approx \boldsymbol{z}_0$). It turns out that the DDIM equation can be used to operate on $\boldsymbol{z}_{t_{\text{step}}}$ for different times $t_{\text{step}}$, which allows us to express the multi-step consistency loss as:

$$|| \text{DDIM}_{t \to t_{\text{step}}}(f(\boldsymbol{z}_t, t), \boldsymbol{z}_t) - \hat{\boldsymbol{z}}_{\text{ref}, t_{\text{step}}}||, \tag{6}$$

where $\hat{\boldsymbol{z}}_{\text{ref}, t_{\text{step}}} = \text{DDIM}_{s \to t_{\text{step}}}(\text{nograd} f(\boldsymbol{z}_s, s), \boldsymbol{z}_s)$ and where the teaching step $\boldsymbol{z}_s = \text{aDDIM}_{t \to s}(x, \boldsymbol{z}_t)$ is an approximation of the probability flow ODE. For now it suffices to think of aDDIM as DDIM. It will be described in detail in section 3.2. In fact, one can drop-in any deterministic sampler (or integrator) in place of aDDIM in the case of *distillation*.

A model can be trained directly on this loss in $z$ space, however make the loss more interpretable and relate it more closely to standard diffusion, we re-parametrize the loss to $x$-space using:

$$||\hat{\boldsymbol{x}}_{\text{diff}}|| = ||f(\boldsymbol{z}_t, t) - \text{invDDIM}_{t \to t_{\text{step}}}(\hat{\boldsymbol{z}}_{\text{ref}, t_{\text{step}}}, \boldsymbol{z}_t)||. \tag{7}$$

This allows the usage of existing losses from diffusion literature, where we have opted for $v$-loss (equivalent to $\text{SNR} + 1$ weighting) because of its prior success in distillation (Salimans & Ho, 2022).

As noted in (Song et al., 2023), consistency in itself is not sufficient to distil a path (always predicting 0 is consistent) and one needs to ensure that the model cannot collapse to these degenerate solutions. Indeed, in our specification observe that when $s = t_{\text{step}}$ then $\hat{\boldsymbol{z}}_{\text{ref}, t_{\text{step}}} = \text{DDIM}_{s \to t_{\text{step}}}(\boldsymbol{z}_s, \hat{\boldsymbol{x}}) = \boldsymbol{z}_s$. As such, the loss of the final step cannot be degenerate because it is equal to:

$$||f(\boldsymbol{z}_t, t) - \text{invDDIM}_{t \to s}(\boldsymbol{z}_s, \boldsymbol{z}_t)||. \tag{8}$$

**Many-step CT is equivalent to Diffusion training** Consistency training learns to integrate the probability flow through time, whereas standard diffusion models learn a path guided by an expectation $\hat{\boldsymbol{x}} = \mathbb{E}[\boldsymbol{x}|\boldsymbol{z}_t]$ that necessarily has to change over time for non-trivial distributions. There are two simple reasons that for many student steps, Multistep CT converges to a diffusion model. 1) At the beginning of a step (specifically $t = t_{\text{step}} + \frac{1}{T}$) the objectives are identical. Secondly, 2) when the number of student steps equals the number

of teacher steps $T$, then every step is equal to the diffusion objective. This can be observed by studying Algorithm 1: let $t = t_{\text{step}} + 1/T$. For consistency *training*, aDDIM reduces to DDIM and observe that in this case $s = t_{\text{step}}$. Hence, under a well-defined model $f$ (such as a $v$-prediction one) $\text{DDIM}_{s \to t_{\text{step}}}$ does nothing and simply produces $\hat{\boldsymbol{z}}_{\text{ref},t_{\text{step}}} = \boldsymbol{z}_s$. Also observe that $\hat{\boldsymbol{z}}_{t_{\text{step}}} = \hat{\boldsymbol{z}}_s$. Further simplification yields:

$$w(t)||\boldsymbol{x}_{\text{diff}}|| = w(t)||\text{invDDIM}_{t \to s}(\boldsymbol{z}_s, \boldsymbol{z}_t) - \hat{\boldsymbol{x}}|| = w(t)||\boldsymbol{x} - \hat{\boldsymbol{x}}|| \tag{9}$$

Where $||\boldsymbol{x} - \hat{\boldsymbol{x}}||$ is the distance between the true datapoint and the model prediction weighted by $w(t)$, which is typical for standard diffusion. Interestingly, in (Song & Dhariwal, 2023) it was found that Euclidean ($\ell_2$) distances typically work better than for consistency models than the more usual squared Euclidean distances ($\ell_2$ squared). We followed their approach because it tended to work better especially for smaller number of student steps, which is a deviation from standard diffusion. Because multistep consistency models tend towards diffusion models, we can state two important hypotheses:

1. *Finetuning Multistep CMs from a pretrained diffusion checkpoint will lead to quicker and more stable convergence.*

2. *As the number of student steps increases, Multistep CMs will rival diffusion model performance, giving a direct trade-off between sample quality and duration.*

Note that this trade-off requires training a new Multistep CM for each of the desired student steps, but given that one starts from a pretrained model, one expects that finetuning requires a fraction of the original training budget.

**What about training in continuous time?** Diffusion models can be easily trained in continuous time by sampling $t \sim \mathcal{U}(0, 1)$, but in Algorithm 1 we have taken the trouble to define $t$ as a discrete grid on $[0, 1]$. One might ask, why not let $t$ be continuously valued. This is certainly

---

**Algorithm 2** Sampling from Multistep CMs

Sample $\boldsymbol{z}_1 \sim \mathcal{N}(0, \mathbf{I})$, $T = \text{student\_steps}$
**for** $t$ in $(\frac{T}{T}, \ldots, \frac{1}{T})$ where $s = t - \frac{1}{T}$ **do**
  $\boldsymbol{z}_s = \text{DDIM}_{t \to s}(f(\boldsymbol{z}_t, t), \boldsymbol{z}_t)$
**end for**

---

possible, *if* the model $f$ would take in an additional conditioning signal to denote in which step it is. This is important because its prediction has to discontinuously change between $t \geq t_{\text{step}}$ (this step) and $t < t_{\text{step}}$ (the next step). In practice, we often train Multistep Consistency Models starting from pre-trained with standard diffusion models, and so having the same interface to the model is simpler. In early experiments we did find this approach to work comparably.

## 3.2 The Adjusted DDIM (aDDIM) sampler.

Popular methods for distilling diffusion models, including the method we propose here, rely on deterministic sampling through numerical integration of the probability flow ODE. In practice, numerical integration of this ODE in a finite number of teacher steps incurs error. For the DDIM integrator (Song et al., 2021a) used for distilling diffusion models in both consistency distillation (Song et al., 2023) and progressive distillation (Salimans & Ho, 2022; Meng et al., 2022) this integration error causes samples to become blurry. To see this quantitatively, consider a hypothetical perfect sampler that first samples $\boldsymbol{x}^* \sim p(\boldsymbol{x}|\boldsymbol{z}_t)$, and then samples $\boldsymbol{z}_s$ using

---

**Algorithm 3** Generating Samples with aDDIM

For all $t$, precompute $x_{\text{var},t} = \eta ||\boldsymbol{x} - \hat{\boldsymbol{x}}(\boldsymbol{z}_t)||^2/d$, or set $x_{\text{var},t} = 0.1/(2 + \alpha_t^2/\sigma_t^2)$.
Sample $\boldsymbol{z}_T \sim \mathcal{N}(0, \mathbf{I})$, choose $\eta \in (0, 1)$
**for** $t$ in $(\frac{T}{T}, \ldots, \frac{1}{T})$ where $s = t - 1/T$ **do**
  $\hat{\boldsymbol{x}} = f(\boldsymbol{z}_t, t)$
  $\hat{\boldsymbol{\epsilon}} = (\boldsymbol{z}_t - \alpha_t \hat{\boldsymbol{x}})/\sigma_t$
  $z_{s,\text{var}} = (\alpha_s - \alpha_t \sigma_s/\sigma_t)^2 \cdot x_{\text{var},t}$
  $\boldsymbol{z}_s = \alpha_s \hat{\boldsymbol{x}} + \sqrt{\sigma_s^2 + (d/||\hat{\boldsymbol{\epsilon}}||^2)z_{s,\text{var}}} \cdot \hat{\boldsymbol{\epsilon}}$
**end for**

---

$$\boldsymbol{z}_s^* = \alpha_s \boldsymbol{x}^* + \sigma_s \frac{\boldsymbol{z}_t - \alpha_t \boldsymbol{x}^*}{\sigma_t} = (\alpha_s - \frac{\alpha_t \sigma_s}{\sigma_t})\boldsymbol{x}^* + \frac{\sigma_s}{\sigma_t}\boldsymbol{z}_t. \tag{10}$$

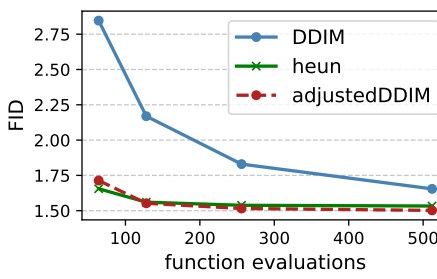 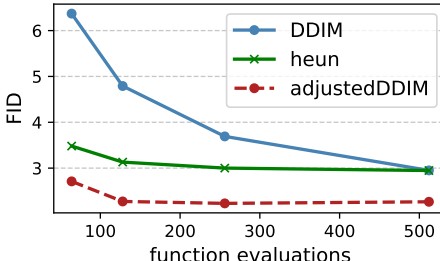

Figure 3: Comparison of sampling methods for the small ImageNet 64 (left) and ImageNet 128 (right) models without distillation. The Heun (with sampler adds a second order correction to DDIM.

If the initial $\boldsymbol{z}_t$ is from the correct distribution $p(\boldsymbol{z}_t)$, the sampled $\boldsymbol{z}_s^*$ would then also be exactly correct. Instead, the DDIM integrator uses

$$\boldsymbol{z}_s^{\mathrm{DDIM}} = (\alpha_s - \alpha_t \sigma_s / \sigma_t)\hat{\boldsymbol{x}} + (\sigma_s / \sigma_t)\boldsymbol{z}_t, \tag{11}$$

with model prediction $\hat{\boldsymbol{x}}$. If $\hat{\boldsymbol{x}} = \mathbb{E}[\boldsymbol{x}|\boldsymbol{z}_t]$, we then have that

$$\mathbb{E}\big[||\boldsymbol{z}_s^*||^2 - ||\boldsymbol{z}_s^{\mathrm{DDIM}}||^2\big|\boldsymbol{z}_t\big] = \mathrm{trace}(\mathrm{Var}[\boldsymbol{z}_s|\boldsymbol{z}_t]), \tag{12}$$

where $\mathrm{Var}[\boldsymbol{z}_s|\boldsymbol{z}_t]$ is the conditional variance of $\boldsymbol{z}_s$ given by

$$\mathrm{Var}[\boldsymbol{z}_s|\boldsymbol{z}_t] = (\alpha_s - \alpha_t \sigma_s / \sigma_t)^2 \cdot \mathrm{Var}[\boldsymbol{x}|\boldsymbol{z}_t], \tag{13}$$

and where $\mathrm{Var}[\boldsymbol{x}|\boldsymbol{z}_t]$ in turn is the variance of $p(\boldsymbol{x}|\boldsymbol{z}_t)$.

The norm of the DDIM iterates is thus too small, reflecting the lack of noise addition in the sampling algorithm. Alternatively, we could say that the model prediction $\hat{\boldsymbol{x}} \approx \mathbb{E}[\boldsymbol{x}|\boldsymbol{z}_t]$ is too smooth.

Currently, the best sample quality is achieved with stochastic samplers, which can be tuned to add exactly enough noise to undo the oversmoothing caused by numerical integration. However, current distillation methods are not well suited to distilling these stochastic samplers directly. Alternatively, deterministic 2$^{\mathrm{nd}}$ order samplers are also not ideal, as they require an additional forward pass during distillation.

Here we therefore propose a new deterministic sampler that aims to achieve the norm increasing effect of noise addition in a deterministic way, *with a single evaluation*. It turns out we can do this by making a simple adjustment to the DDIM sampler, and we therefore call our new method Adjusted DDIM (aDDIM). Our modification is heuristic and is not more theoretically justified than the original DDIM sampler. However, empirically we find aDDIM to work very well leading to improved FID scores (Fig. 3) and thus a stronger deterministic teacher.

aDDIM performs on par with the 2nd order Heun sampler on Imagenet64 and outperforms it on Imagenet128. Indicating that a noise correction works just as well or better than a 2$^{\mathrm{nd}}$ order correction. Interestingly, we also found that the 2$^{\mathrm{nd}}$ order Heun sampler (Karras et al., 2022) only works well with the noise schedule introduced in the same work (see App. A.4 for more details).

Instead of adding noise to our sampled $\boldsymbol{z}_s$, we simply increase the contribution of our deterministic estimate of the noise $\hat{\boldsymbol{\epsilon}} = (\boldsymbol{z}_t - \alpha_t \hat{\boldsymbol{x}})/\sigma_t$. Assuming that $\hat{\boldsymbol{x}}$ and $\hat{\boldsymbol{\epsilon}}$ are orthogonal, we achieve the correct norm for our sampling iterates using:

$$\boldsymbol{z}_s^{\mathrm{aDDIM}} = \alpha_s \hat{\boldsymbol{x}} + \sqrt{\sigma_s^2 + \mathrm{tr}(\mathrm{Var}[\boldsymbol{z}_s|\boldsymbol{z}_t])/||\hat{\boldsymbol{\epsilon}}||^2} \cdot \hat{\boldsymbol{\epsilon}}. \tag{14}$$

In practice, we can estimate $\mathrm{tr}(\mathrm{Var}[\boldsymbol{z}_s|\boldsymbol{z}_t]) = (\alpha_s - \alpha_t \sigma_s / \sigma_t)^2 \cdot \mathrm{tr}(\mathrm{Var}[\boldsymbol{x}|\boldsymbol{z}_t])$ empirically on the data by computing beforehand $\mathrm{tr}(\mathrm{Var}[\boldsymbol{x}|\boldsymbol{z}_t]) = \eta||\hat{\boldsymbol{x}}(\boldsymbol{z}_t) - \boldsymbol{x}||^2$ for all relevant timesteps $t$. Here $\eta$ is a hyperparameter which we set to 0.75. Alternatively, we obtain equally good results

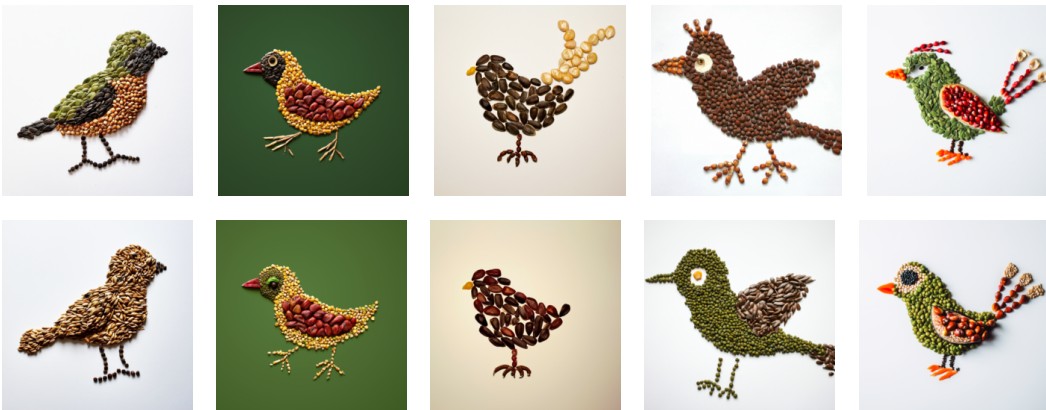

Figure 4: Another qualititative comparison between a multistep consistency and teacher, using the same prompt. Top: ours, a distilled 16-step concistency model (3.2 secs). Bottom: generated samples using a 100-step DDIM diffusion model (39 secs). Both models use the same initial noise.

by approximating the posterior variance analytically with $\text{tr}(\text{Var}[\boldsymbol{x}|\boldsymbol{z}_t])/d = 0.1/(2 + \alpha_t^2/\sigma_t^2)$, for data dimension $d$, which can be interpreted as 10% of the posterior variance of $\boldsymbol{x}$ if its prior was factorized Gaussian with variance of 0.5. In either case, note that $\text{Var}[\boldsymbol{z}_s|\boldsymbol{z}_t]$ vanishes as $s \to t$: in the many-step limit the aDDIM update thus becomes identical to the original DDIM update. For a complete description see Algorithm 3.

Note that aDDIM only replaced the teacher steps. The student model uses a vanilla DDIM step which learns to predict the trajectory of the teacher with aDDIM. The student DDIM step only serves as a convenient output parameterization, and the student could just as well predict $z_s$ directly.

## 4 RELATED WORK

Multistep Consistency Models are a direct combination of (Song et al., 2023; Song & Dhariwal, 2023) and TRACT (Berthelot et al., 2023). Compared to consistency models, we propose to operate on multiple stages, which simplifies the modelling task and improves performance significantly. On the other hand, TRACT limits itself to distillation and uses the self-evaluation from consistency models to distill models over multiple stages. The stages are progressively reduced to either one or two stages and thus steps. The end-goal of TRACT is again to sample in either one or two steps, whereas we believe better results can be obtained by optimizing for a slightly larger number of steps. We show that this more conservative target, in combination with our improved sampler and annealed schedule, leads to significant improvements in terms of image quality that closes the gap between sample quality of standard diffusion and low-step diffusion-inspired approaches.

Earlier, DDIM (Song et al., 2021a) showed that deterministic samplers degrade more gracefully than the stochastic sampler used by Ho et al. (2020) when limiting the number of sampling steps. Karras et al. (2022) proposed a second order Heun sampler to reduce the number of steps (and function evaluations), while Jolicoeur-Martineau et al. (2021) studied different SDE integrators to reduce function evaluations. Progressive Distillation (Salimans & Ho, 2022; Meng et al., 2022) distills diffusion models in stages, which limits the number of model evaluations during training while exponentially reducing the required number of sampling steps with the number stages.

Other methods inspired by diffusion such as Rectified Flows (Liu et al., 2023a) and Flow Matching (Lipman et al., 2023) have also tried to reduce sampling times. In practice however, flow matching and rectified flows are generally used to map to a standard normal distribution and reduce to standard diffusion. As a consequence, on its own they still require many evaluation steps. In Rectified Flows, a distillation approach is proposed that does reduce sampling steps more significantly, but this comes at the expense of sample quality.

Table 1: Imagenet performance with multistep consistency training (CT) and consistency distillation (CD), started from a pretrained diffusion model. A baseline with the aDDIM sampler on the base model is included.

| | | | Small | | | | Large | | | |
| | | | ImageNet64 | | ImageNet128 | | ImageNet64 | | ImageNet128 | |
| | | Steps | Train | Distill | Train | Distill | Train | Distill | Train | Distill |
|---|---|---|---|---|---|---|---|---|---|---|
| Base | Consistency Model | 1 | 7.2 | 4.3 | 16.0 | 8.5 | 6.4 | 3.2 | 14.5 | 7.0 |
| | MultiStep CM (ours) | 2 | 2.7 | 2.0 | 6.0 | 3.1 | 2.3 | 1.9 | 4.2 | 3.1 |
| | MultiStep CM (ours) | 4 | 1.8 | 1.7 | 4.0 | 2.4 | 1.6 | 1.6 | 2.7 | 2.3 |
| | MultiStep CM (ours) | 8 | 1.5 | 1.6 | 3.3 | 2.1 | 1.5 | 1.4 | 2.2 | 2.1 |
| | MultiStep CM (ours) | 16 | 1.5 | 1.5 | 3.4 | 2.0 | 1.6 | 1.4 | 2.3 | 2.0 |
| | Diffusion (aDDIM) | 512 | 1.5 | | 2.2 | | 1.4 | | 2.2 | |

Table 2: Ablation of CD on Image128 with and without annealing the teacher steps on ImageNet128. Annealing the teacher stepsize improves the performance.

| Steps | $(64 \to 1280)$ | (step = 128) | (step = 256) | (step = 1024) |
|---|---|---|---|---|
| 1 | **7.0** | 8.8 | 7.6 | 10.8 |
| 2 | **3.1** | 5.3 | 3.6 | 3.8 |
| 4 | **2.3** | 5.0 | 3.5 | 2.6 |
| 8 | **2.1** | 4.9 | 3.2 | 2.2 |

Table 3: Comparison between PD (Salimans & Ho, 2022) and CT/CD on ImageNet64 on the small model.

| Steps | CT (ours) | CD (ours) | PD |
|---|---|---|---|
| 1 | 7.2 | 4.3 | 10.7 |
| 2 | 2.7 | 2.0 | 4.7 |
| 4 | 1.8 | 1.7 | 2.4 |
| 8 | 1.5 | 1.6 | 1.8 |

**Adversarial distillation** Distillation to a few steps was very difficult to do using only simple distance metrics. Therefore, many works resort to a form of adversarial training. For example Luo et al. (2023) distill the knowledge from the diffusion model into a single-step model and Zheng et al. (2023) use specialized architectures to distill the ODE trajectory from a pre-created noise-sample pair dataset. A very similar approach to ours is Consistency Trajectory Models (CTMs) (Kim et al., 2023), which are trained to arbitrarily integrate to a given timestep. This is implemented by modifying the inputs of the denoising network to include an endpoint of the integration. Although CTMs produce very high quality image samples in a few steps, their performance relies on adversarial training: Without it, CTMs cannot produce great samples and have a considerable gap in FID score. In contrast, our Multistep CMs can be trained with simple distance metrics and still achieve very good FID scores under a few sampling steps. A possible explanation is that it is much easier to learn a handful of fixed integration trajectories (Multistep CMs) instead of every possible integration with arbitrary endpoints (CTMs). Another advantage of Multistep CMs is that the inputs to the denoising network are not changed, making fine-tuning of existing diffusion models to Multistep CMs very straightforward.

## 5 EXPERIMENTS

Our experiments focus on a quantitative comparison using the FID score on ImageNet as well as a qualitative assessment on large scale Text-to-Image models. These experiments should make our approach comparable to existing academic work while also giving insight in how multi-step distillation works at scale.

### 5.1 EVALUATION ON IMAGENET

For our ImageNet experiments we trained diffusion models on ImageNet64 and ImageNet128 in a base and large variant. We initialize the consistency models from the pre-trained diffusion model weights which we found to greatly increase robustness and convergence. Both consistency training and distillation are used. Classifier Free Guidance (Ho & Salimans, 2022) was used only on the base ImageNet128 experiments. For all other experiments we did not use guidance because

Table 4: Ablation of the aDDIM teacher on ImageNet64.

| Student Steps | DDIM | aDDIM |
|---|---|---|
| 1 | **3.91** | 4.35 |
| 2 | **1.99** | 2.02 |
| 4 | 1.77 | **1.68** |
| 8 | 1.70 | **1.58** |
| 16 | 1.72 | **1.54** |

it did not significantly improve the FID scores of the
diffusion model. All consistency models are trained for 200,000 steps with a batch size of 2048 and a teacher step schedule that anneals from 64 to 1280 in 100.000 train steps with an exponential schedule.

In Table 1 the performance improves when the student step count increases. There are generally two patterns we observe: As the student steps increase, performance improves. This validates our hypothesis that more student steps are a useful trade-off between sample quality and speed. Conveniently, this happens very early: even on a complicated dataset such as ImageNet128, our base model variant is able to achieve 2.1 FID with just 8 student steps.

To draw a direct comparison between Progressive Distillation (PD) (Salimans & Ho, 2022) and our approaches, we reimplement PD using aDDIM and we use same base architecture, as reported in Table 3. With our improvements, PD can attain better performance than previously reported in literature. However, compared to MultiStep CT and CD it starts to degrade in sample quality at low step counts. For instance, a 4-step PD model attains an FID of 2.4 whereas CD achieves 1.7.

In Tbl. 4 we ablate the effect of using adjusted DDIM as a teacher. Empirically, we observe that the adjusted sampler is important when more student steps are used. In contrast, vanilla DDIM works better when few steps are taken and the student does not get close to the teacher as measured in FID.

Further we ablate whether annealing the step schedule is important to attain good performance. As can be seen in Tbl. 2, it is especially important for low multistep models to anneal the schedule. In these experiments, annealing always achieves better performance than tests with constant teacher steps at $128, 256, 1024$. As more student steps are taken, the importance of the annealing schedule decreases.

**Literature Comparison** Compared to existing works in literature, we achieve SOTA FID scores in both ImageNet64 and Imagenet128 with 4-step and 8-step generation. Interestingly, we achieve approximately the same performance using single step CD compared to iCT-deep (Song & Dhariwal, 2023), which achieves this result using direct consistency training. Since direct training has been empirically shown to be a more difficult task, one could conclude that some of our hyperparameter choices may still be suboptimal in the extreme low-step regime. Conversely, this may also mean that multistep consistency is less sensitive to hyperparameter choices.

In addition, we compare on ImageNet128 to our reimplementation of Progressive Distillation. Unfortunately, ImageNet128 has not been widely adopted as a few-step benchmark, possibly because a working deterministic sampler has been missing until this point. For reference we

Table 5: Literature Comparison on ImageNet.

| Method | NFE | FID | non-adv |
|---|---|---|---|
| *Imagenet 64 x 64* | | | |
| DDIM (Song et al., 2021a) | 10 | 18.7 | ✓ |
| DFNO (LPIPS) (Zheng et al., 2023) | 1 | 7.83 | ✓ |
| TRACT (Berthelot et al., 2023) | 1 | 7.43 | ✓ |
| | 2 | 4.97 | ✓ |
| | 4 | 2.93 | ✓ |
| | 8 | 2.41 | ✓ |
| Diff-Instruct | 1 | 5.57 | |
| PD (Salimans & Ho, 2022) | 1 | 10.7 | ✓ |
| (reimpl. with aDDIM) | 2 | 4.7 | ✓ |
| | 4 | 2.4 | ✓ |
| | 8 | 1.7 | ✓ |
| PD Stochastic (Meng et al., 2022) | 1 | 18.5 | ✓ |
| | 2 | 5.81 | ✓ |
| | 4 | 2.24 | ✓ |
| | 8 | 2.31 | ✓ |
| CD (LPIPS) (Song et al., 2023) | 1 | 6.20 | ✓ |
| | 2 | 4.70 | ✓ |
| | 3 | 4.32 | ✓ |
| PD (LPIPS) (Song et al., 2023) | 1 | 7.88 | ✓ |
| | 2 | 5.74 | ✓ |
| | 3 | 4.92 | ✓ |
| iCT-deep (Song & Dhariwal, 2023) | 1 | 3.25 | ✓ |
| iCT-deep | 2 | 2.77 | ✓ |
| CTM (Kim et al., 2023) | 1 | **1.9** | |
| | 2 | **1.7** | |
| DMD (Yin et al., 2023) | 1 | 2.6 | |
| MultiStep-CT (ours) | 2 | 2.3 | ✓ |
| | 4 | **1.6** | ✓ |
| | 8 | 1.5 | ✓ |
| MultiStep-CD (ours) | 1 | 3.2 | ✓ |
| | 2 | 1.9 | ✓ |
| | 4 | **1.6** | ✓ |
| | 8 | **1.4** | ✓ |
| *Imagenet 128 x 128* | | | |
| VDM++ (Kingma & Gao, 2023) | 512 | 1.75 | ✓ |
| PD (Salimans & Ho, 2022) | 2 | 8.0 | ✓ |
| (reimpl. with aDDIM) | 4 | 3.8 | ✓ |
| | 8 | 2.5 | ✓ |
| MultiStep-CT (ours) | 2 | 4.2 | ✓ |
| | 4 | 2.7 | ✓ |
| | 8 | 2.2 | ✓ |
| MultiStep-CD (ours) | 2 | **3.1** | ✓ |
| | 4 | **2.3** | ✓ |
| | 8 | **2.1** | ✓ |

Table 6: Text to Image performance. Note that when 8/16-step Consistency is compared to a teacher model that is only guidance distilled at 256 steps, there is practically no performance loss.

| Method | NFE | $FID_{30k}$ | $FID_{5k}$ | CLIP | non-adv |
|---|---|---|---|---|---|
| SDv1.5 (Rombach et al., 2022) low g (from DMD) | 512 | 8.8 | | - | ✓ |
| high g (from DMD) | 512 | 13.5 | | **0.322** | ✓ |
| DMD (low guidance) (Yin et al., 2023) | 1 | 11.5 | | - | |
| (high guidance) | 1 | 14.9 | | 0.32 | |
| UFOGen (Xu et al., 2023) | 1 | 12.8 | 22.5 | 0.311 | |
| | 4 | | 22.1 | 0.307 | |
| InstaFlow-1.7B (Liu et al., 2023b) | 1 | 11.8 | 22.4 | 0.309 | ✓ |
| PeRFlow (Yan et al., 2024) | 4 | 11.3 | | | ✓ |
| Teacher Diffusion Model g=0.5 (ddpm) | 256 | **7.9** | **13.6** | 0.305 | ✓ |
| guidance distilled (ddim) | 256 | 8.2 | 13.8 | 0.300 | ✓ |
| Multistep-CD (teacher g=0.5) | 4 | 8.7 | 14.4 | 0.298 | ✓ |
| | 8 | 8.1 | 13.8 | 0.300 | ✓ |
| | 16 | **7.9** | 13.9 | 0.300 | ✓ |
| Teacher Diffusion Model g=3 (ddpm) | 256 | 12.7 | 18.1 | 0.315 | ✓ |
| guidance distilled (ddim) | 256 | 13.9 | 19.0 | 0.312 | ✓ |
| Multistep-CD (teacher g=3) | 4 | 12.4 | 18.1 | 0.311 | ✓ |
| | 8 | 13.9 | 19.6 | 0.311 | ✓ |
| | 16 | 14.4 | 20.0 | 0.312 | ✓ |

also provide the recent result from (Kingma & Gao, 2023). Further, with these results we hope to put ImageNet128 on the map for few-step diffusion model evaluation.

## 5.2 Evaluation on Text to Image modelling

In addition to the analysis on ImageNet, we study the effects on text-to-image models. We distill a 16-step consistency model from a base teacher model. In Table 6 one can see that Multistep CD is able to distill its teacher almost perfectly in terms of FID. The loss of clip score can be attributed to the guidance distillation, which a baseline 256-step student model also has trouble distilling. Compared to the guidance-distilled baseline, the 16-CD model has no loss in performance measured in CLIP and FID on the low guidance setting (and for the high guidance setting only a minor degradation in FID). Even the 8-step CD model attains an impressive FID score of 8.1, which is well below the existing literature.

In Figure 2 and 6 we compare samples from our 16-step CD aDDIM distilled model to the original 100-step DDIM sampler. Because the random seed is shared we can easily compare the samples between these models, and we can see that there are generally minor differences. In our own experience, we often find certain details more precise, at a slight cost of overall construction. Another comparison in Figure 4 shows the difference between a DDIM distilled model (equivalent to $\eta = 0$ in aDDIM) and the standard DDIM sampler. Again we see many similarities when sharing the same initial random seed.

## 6 Conclusions

In conclusion, this paper presents Multistep Consistency Models, a simple unification between Consistency Models (Song et al., 2023) and TRACT (Berthelot et al., 2023) that closes the performance gap between standard diffusion and few-step sampling. Multistep Consistency gives a direct trade-off between sample quality and speed, achieving performance comparable to standard diffusion in as little as eight steps. The main limitation of multistep consistency is that one pays the price of several function evaluations to generate a sample. Here, adversarial approaches generally perform better when only one or two evaluations are permitted, but they come the cost of more difficult training dynamics.

**Broader Impacts**  This paper proposes a method to speed up sampling from diffusion models. Although generative models may be used for positive applications such as enhancing human creativity or drug discovery, they may also be used to create deepfakes or misinformation. Hence, enabling faster sampling may amplify both the positive and the negative impacts of generative modelling.

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

## A  Experimental Details

### A.1  Setup

In this paper we follow the setup from simple diffusion (Hoogeboom et al., 2023). Following their approach, we use a standard UViTs. These are UNets with MLP blocks instead of convolutional layers when a block has self-attention, making the entire block a transformer block. This contains the details for the architecture and how to define diffusion process. There are some minor specifics which we share per experiment below. *All runs are initialized using the parameters of a pretrained diffusion models.*

**Multistep Consistency Hyperparameters**  For all ImageNet runs (small/large, 1 through 16 step) we use a log-linear interpolated schedule from 64 teacher steps to 1280 teacher steps, annealed over 100000 training iterations which means $N_{\text{teacher}}(i) = \exp(\log 64 + \text{clip}(i/100.000, 0, 1) \cdot (\log 1280 - \log 64))$. The batch size is 2048. We use a `xvar_frac` of 0.75 for aDDIM. And we use a huber epsilon of 1e-4. The model is trained for 200000 steps. The interpolation starts quite low and takes a long time, and these settings are somewhat excessive for the larger student step models such as the 8- or 16-step model. However, fixing these settings for the model allowed for clean comparisons. These runs anneal the teacher steps using

For the text-to-image model, we ran consistency distillation where we kept the teacher steps fixed at 256 and used an `xvar_frac` of 0.75. Note that the `xvar_frac` should always be computed on the conditional output, not the guided output (so guidance zero). We used a huber epsilon of 1, which is essentially a scalar-scaled l2 squared loss for the normalized [-1, 1] domain of interest. We train these models for 30000 steps at a batch size of 2048.

**ImageNet64**  For the ImageNet64 experiments, the levels of the UViT small are as follows. Down: 3 ResNet blocks with 256 channels, 3 Transformer Blocks with 512 channels both stages ending with an average pool. Middle: 16 transformer blocks with 1024 channels, mlp expansion factor is 4. Up, matching the down blocks, starting a stage with a nearest neighbour upsampling and obviously no pooling. Dropout is applied to the middle with a factor of 0.2. For the large variant, all channels are multiplied by 2, and dropout is applied to all transformers albeit with a lower factor of 0.1. The network is trained with an interpolated cosine schedule from noise resolution 32 to 64 at a resolution of 64 (this is practically identical to a normal cosine schedule). The small and large model have 394M and 1.23B parameters, respectively.

**ImageNet128**  For the ImageNet128 experiments, the UViT is the same as the UViT for ImageNet64, but with an extra 3 ResNet Blocks at the resolution 128x128 with 128 channels at both the start and the end of the UViT. For completeness, down: 3 ResNet blocks with 128 channels, 3 ResNet blocks with 256 channels, 3 Transformer Blocks with 512 channels both stages ending with an average pool. Middle: 16 transformer blocks with 1024 channels, mlp expansion factor is 4. Up, matching the down blocks, starting a stage with a nearest neighbour upsampling and no pooling. The small and large model have 397M and 1.25B parameters, respectively.

Different from before, dropout is applied to the middle with only a factor of 0.1. For the large variant, all channels are multiplied by 2, and dropout is applied to all *blocks* (both convolutional and transformer) except for the ones at the resolution of 128, also at a factor of 0.1. The network is trained with an interpolated cosine schedule from noise resolution 32 to 128 at a resolution of 128, with a multiscale loss (Hoogeboom et al., 2023) that $2 \times 2$ average pools once.

**Text-to-Image**  The text-to-image model is directly trained on $512 \times 512$, with a multiscale loss and an interpolated cosine schedule starting at noise resolution 32 and ending at 512. The UViT has the following stages, down: 3 ResNet blocks at 128 channels, 3 ResNet blocks at 256 channels, 3 ResNet blocks at 1024 channels, 3 transformer blocks at 2048 channels, average pool at the end of each stage. Mid: 16 transformer blocks with 4096 channels and

dropout ratio 0.1. Up: identical to reversed down with nearest neighbour instead of average pooling.

## A.2 COMPUTE RESOURCES

All small model variants are run on 64 TPUv5e chips. For ImageNet64 CT takes 2.7 training steps per second and CD takes 2.5 steps per sec. For ImageNet128 CT takes 2.2 training steps per second and CD takes 1.7 steps per sec.

The large variants are trained on 256 TPUv5e chips. For ImageNet64 CT takes 2.9 training steps per second and CD takes 2.5 steps per sec. For ImageNet128 CT it takes 2.2 training steps per second and CD takes 1.8 steps per second. The text to image experiment is also run on 256 TPUve chips and takes 0.71 steps per second to train, and is only trained for 30000 iterations.

All models use a batch size of 2048 during training.

## A.3 DATASETS

The models in this paper are trained on ImageNet dataset (Russakovsky et al., 2015). The text to image model is trained on a privately licensed text-to-image dataset, comparable with public text-to-image datasets but filtered for content.

## A.4 TEACHER SAMPLING

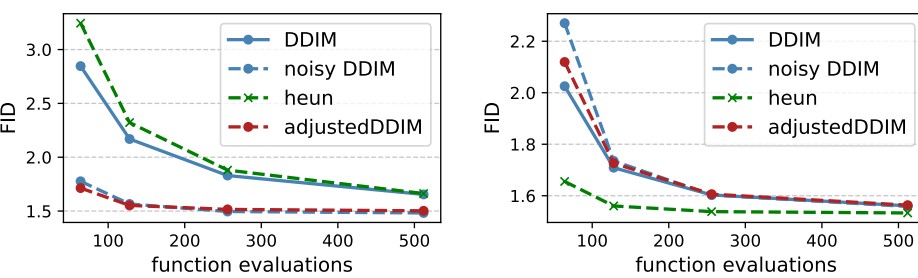

Figure 5: Comparison of different sampling methods for the cosine schedule (left) and the sigma schedule used by Karras et al. (2022) (right) on Imagenet64. *Note that aDDIM with a (shifted) cosine schedule is the best performing model overall except for the 64 function evaluation.*

Fig. 5 compares various samplers including the 2nd order Heun sampler. Additionally, a stochastic version of DDIM is included (noise DDIM) where we add random Guassian noise directly to the model prediction. This direct noise injection breaks the determinism of DDIM and is therefore not a useful sampler for consistent distillation. However, it behaves very similarly to the aDDIM which seems to indicate that our heuristic noise correction is accurately simulating the positive effects of noise injection in the sampler.

Interestingly, we observe a significant difference in the relative quality of various sampling methods depending on the noise schedule used at evaluation. The Heun sampler favors the schedule introduced by Karras (Karras et al., 2022) while the noisy methods seem to work better with a standard cosine schedule. One possible explanation is that the asymptotic behavior of the cosine schedule favours the noise injection methods. Previous work has indicated that the asymptotic behavior a noise schedule is important to fully capture the data distribution (Lin et al., 2024). We consider investigating the interaction between schedules and samplers and interesting opportunity for future work.

## A.5 Additional results

In Figure 6, some additional results are shown for the same prompt. Again, the distilled model is very similar to the original teacher model with minor variations.

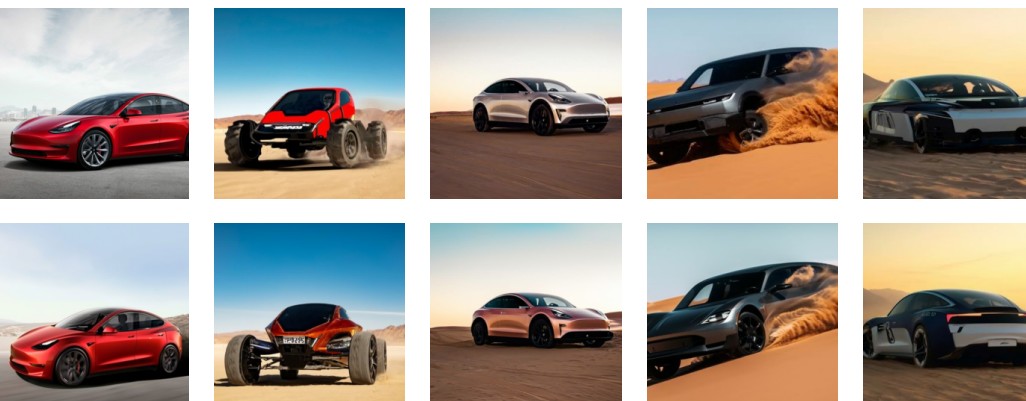

Figure 6: Another qualititative comparison between a multistep consistency and diffusion model. Top: ours, samples from aDDIM distilled 16-step concistency model (3.2 secs). Bottom: generated samples using a 100-step DDIM diffusion model (39 secs). Both models use the same initial noise.

