# OpenReview forum: "Multistep Consistency Models"
_ICLR.cc/2025/Conference — ICLR 2025 Conference Withdrawn Submission_

### Official Review · Reviewer_Njdj · 2024-10-24

**Soundness:** 3
**Presentation:** 1
**Contribution:** 2
**Rating:** 3
**Confidence:** 4

**Summary:**

This paper extends CMs to multiple steps, achieving a balanced trade-off between CM and DM. Additionally, it introduces an adjusted DDIM step designed to reduce variance mismatch, which arises from using an expected value in the DDIM step instead of sampling from the true distribution. The authors evaluate their approach on several datasets, demonstrating its efficiency and effectiveness.

**Strengths:**

1. MCM achieves a trade-off between DM and CM. Unlike CTM, MCM does not rely on GAN loss to achieve high-quality samples, which yields more stable training and more flexible application.
2. The author discussed that the DDIM step tends to underestimate the variance. The proposed aDDIM can address this mismatch and improve the results.
3. The author provides both lower-resolution and higher-resolution results using MCM, with and without distillation. These comprehensive experiments demonstrate the efficiency of MCM.

**Weaknesses:**

1. **(major) the trade-off seems to be limited.** This paper achieves a trade-off between CMs and DMs. However, this trade-off is achieved by set the number of student steps in advance. While this approach demonstrates a clear improvement over standard CMs and DMs in terms of generation quality and speed, respectively, it significantly limits flexibility due to the fixed design choice. The authors also discuss CTMs, arguing that adversarial training is required to ensure high-quality samples. However, I am not entirely convinced that this is a major drawback, as using GANs is a mostly free lunch, and when combined with DSM and CTM loss, the training process is not as unstable compared to standalone GAN training.
Do you think it is possible to train a single model using the MCM objective to achieve a real post-training trade-off?

2. **(major) some design choices are unmotivated.**

(a) the $\eta$ in aDDIM is set to 0.75. How is this chosen?

(b) Alternatively, aDDIM can be set so that the variance can be interpreted as 10% of the posterior variance of x if its prior was factorized Gaussian with variance of 0.5. Why this choice?

(c) "We initialize the consistency models from the pre-trained diffusion model weights which we found to greatly increase robustness and convergence." It is not surprising this will help. But what if there is no such initialization? Considering that CM and CTM can be trained without this initialization, it would be helpful to see the results.



3. **Some results are not convincing.**

(a) "Empirically, we observe that the adjusted sampler is important when more student steps are used. In contrast, vanilla DDIM works better when few steps are taken and the student does not get close to the teacher as measured in FID." This seems to be weird to me.
As far as I understand, aDDIM and DDIM will converge when the number of steps becomes larger. Therefore, I will expect the gain to become smaller when using more steps. ADDIM will be more accurate (in terms of variance) compared to DDIM when the step number is small. However, it seems that the experimental results are the opposite. Any explanation?

(b) This may be a little too much. But I am asking just out of curiosity. In the experiments, you use UViT as the network. However, both CTM and CM use the standard UNet. How does this architectural difference contribute to the performance? Will MCM still perform well when using UNet?


4. **Missing related works**. The motivation behind aDDIM is to address the underestimation of variance in the DDIM step. To remedy this, aDDIM increases the predicted noise contribution to match the empirical variance. Similar ideas have been proposed in [1][2] (and also [2] gives a good summary of other approaches in its related work section), though a key distinction is that aDDIM remains deterministic, whereas these methods still involve adding noise. However, given the shared rationale between aDDIM and these approaches, I suggest discussing them in related works.


5. **(minor) writing and formatting is sometimes confusing and contains typos.** Here are some examples:

(a) Algorithm 1. There should be punctuation after the first line. I thought $N_{per segment}$ was part of the first line for a while.

(b)  line 215: I do not think $T$ is clearly defined somewhere?

(c) line 225-226, typo: ...Euclidean distances typically work better than for consistency models than the more usual squared Euclidean distances...

(d) references for tables are not consistent. Line 454/463 use *Tbl*.XXX, but line 446/437 use *Table* XXX.



[1] Bao, Fan, et al. "Estimating the Optimal Covariance with Imperfect Mean in Diffusion Probabilistic Models." ICML 2022.
[2] Ou, Zijing, et al. "Diffusion Model With Optimal Covariance Matching." arXiv.

**Questions:**

1. I have one question regarding the objective of the network.
Imagine $t$ is one of the boundaries for the network to switch targets. Assume $t_1 = t-\delta$ and $t_2=t+\delta$.
We expect the network to predict two different targets for $t_1$ and $t_2$. Is this right?
So how does the network handle this sudden change? Will this introduce higher variance or instability to the training process?
Or will these two targets coincide under some assumption?

---

### Official Review · Reviewer_nKt8 · 2024-10-28

**Soundness:** 2
**Presentation:** 2
**Contribution:** 1
**Rating:** 3
**Confidence:** 4

**Summary:**

This paper introduces a segmented distillation method, along with aDDIM.

**Strengths:**

This paper proposes a method that can be directly applied to the large-scale models like SDXL.

**Weaknesses:**

Major Comments

- In line 135, what does the term "difficult" refer to? Is it about the instability during training or the model’s overall performance?
- In the paragraph starting at line 145, my understanding is that Consistency Models did not originally introduce a gap between model evaluations at t and s, although extending in that direction seems intuitive. Additionally, even as s->t, would longer propagation during training significantly affect the author's setup? To my knowledge, 200k iterations with an EMA rate of 0.9999 are sufficient for convergence, regardless of propagation length.
- In line 176, the phrase "difficult to learn when it jumps between modes" feels inappropriate. Simply stating that "jumping between modes represents a more complex function" seems sufficient. If this is why we need MCM, wouldn’t increasing the model’s capacity address the issue?
- In line 182, there are already several papers that handle multiple segments. What makes this approach novel?
- In line 188, would it be better to set $z_{t_{step}}$ as the target instead of x? What’s the intuition here? Moreover, transitioning from Eq. (6) to Eq. (7) with differences in the x-space appears contradictory.
- Could the model be trained using LPIPS? What would happen if LPIPS were used instead? Would it lead to better performance with smaller steps? Since using l2 instead of squared l2 no longer ensures the denoiser function as the optimal solution, I suspect LPIPS could be beneficial if denoiser optimality is not the goal.
- In the paragraph starting at line 240, I’m confused. In Consistency Models, performance in continuous time was poor. How does the proposed algorithm work better in continuous time? Could you provide more intuition?
- I found Section 3.2 on aDDIM unclear. Given that there are already several papers covering segmented timestep distillation (the claimed contribution), I only see aDDIM as the novel part. However, since I don’t fully understand this concept, I can't recommend acceptance. Please make it clearer.
- Even if aDDIM is clear, Table 4 shows only marginal effects of aDDIM. Can the authors provide further arguments for why aDDIM deserves top-tier conference recognition?
Regarding the loss function, given the existing GAN-based distillation approaches like ADD [1] or LADD [2], wouldn’t it be better to adopt an adversarial loss? Why insist on l2?
- In line 265, what does "this integration error causes samples to become blurry" mean? Which samples? Multi-step or one-step samples? What exactly is the integration error? If DDIM works, how does integration error affect sample quality?
- A more comprehensive related work section on distillation models is needed. For instance, [3] is the first work to distill diffusion models. [4] is also closely related, particularly in the DDPM setup.
- Could the proposed distillation method works on DiT?
- I’m curious how the proposed algorithm fits into both the VE and FM settings. Could the authors discuss this in the main paper?

Minor Comments

- In line 99, Song's Score SDE paper didn’t describe the object as a denoiser. The concept of denoising was introduced later. Song's derivation of score matching was combined with the Tweedie formula to reach that conclusion. Please cite either the Tweedie formula or other papers that clearly mention this. I recommend [5] if you prefer to stay within the diffusion community.
- Make $z_{t_{step}}$ in line 188 bold.
- Change ADDIM in the heading of Section 3.2 to aDDIM.

[1] Sauer, Axel, et al. "Adversarial diffusion distillation." arXiv preprint arXiv:2311.17042 (2023).
[2] Sauer, Axel, et al. "Fast high-resolution image synthesis with latent adversarial diffusion distillation." arXiv preprint arXiv:2403.12015 (2024).
[3] Luhman, Eric, and Troy Luhman. "Knowledge distillation in iterative generative models for improved sampling speed." arXiv preprint arXiv:2101.02388 (2021).
[4] Zheng, Jianbin, et al. "Trajectory consistency distillation." arXiv preprint arXiv:2402.19159 (2024).
[5] Jolicoeur-Martineau, Alexia, et al. "Adversarial score matching and improved sampling for image generation." arXiv preprint arXiv:2009.05475 (2020).

**Questions:**

-

---

### Official Review · Reviewer_WzCU · 2024-11-02

**Soundness:** 2
**Presentation:** 3
**Contribution:** 2
**Rating:** 5
**Confidence:** 4

**Summary:**

This work extends the Consistency Model by leveraging the inversion of DDIM to generate a clean regression target based on the DDIM solution at a middle timestep, starting from a noisier timestep. The proposed method demonstrates strong empirical performance in diffusion distillation without the need for adversarial training.

**Strengths:**

This work extends the Consistency Model by leveraging the inversion of DDIM to generate a clean regression target based on the DDIM solution at a middle timestep, starting from a noisier timestep. The proposed method demonstrates strong empirical performance in diffusion distillation without the need for adversarial training.

**Weaknesses:**

1. There are some typos, such as in line 101 where "*One than...*" should be "*One then...*". Additionally, some characters should be in bold but are not. Some parts are not consistent, such as using both *Table XX* and *Tbl. XX*.

2. Regarding the flow in Section 2, would it improve clarity to switch the paragraphs *Consistency Training and Distillation* and *DDIM Sampler*? Currently, the notation $\mathrm{DDIM}_{t\rightarrow s}$ appears in Eq. (3) without prior introduction. Additionally, certain terms need definition or mention, such as $\mathbf{x}$-teacher. I appreciate the strong performance of the method, as highlighted in the abstract and introduction. However, further elaboration on the proposed method and the sources of its performance in these sections would be greatly appreciated.

3. Could you clarify the fundamental difference between the proposed method and [1], particularly regarding the design of the distillation loss and model parameterizations? [1] also unifies the distillation-based model (which involves a few long jump steps for generation) and the diffusion model (which uses many steps for generation). The proposal seems to be a minor modification of this framework.


4. In Table 1, is the Consistency Model's result also from UViTs backbones? In Table 5, is the proposed method using standard UNet structures that baselines were using? I'd like to know if all the comparisons are fair in terms of architectures, model sizes. Additionally, what is the teacher model used in the distillation training? If it is based on EDM [3], that would be fine; otherwise, it may not be an apples-to-apples comparison with respect to the baselines.


5. Could you explain how one can ensure that the learned model, using objective (8) with the adjusted DDIM for teacher steps, converges to either the teacher model (for distillation) or the true model (for training from scratch)—similar to Theorem 1 and Theorem 2 in [2]? The adjusted DDIM seems to alter the learning target.

6. The DDIM-related solutions (e.g., $\mathrm{DDIM}$ or $\mathrm{invDDIM}$, which seem to solve in a single step from $t$ to $s$ even when the gap $|t - s|$ is large) appear to take large step jumps directly, rather than solving in a time-stepping manner—this approach could introduce significant discretization errors during training.

7. Is there evidence supporting the claim that the proposed method is more stable and converges faster (presumably compared to the consistency model)? Additionally, how many function evaluations are required in the proposed method compared to those needed for training the Consistency Model? Overall, does the proposed method still converge faster regardless of model sizes?


[1] Kim, D., Lai, C.H., Liao, W.H., Murata, N., Takida, Y., Uesaka, T., He, Y., Mitsufuji, Y. and Ermon, S., 2023. Consistency trajectory models: Learning probability flow ode trajectory of diffusion.

[2] Song, Y., Dhariwal, P., Chen, M., & Sutskever, I. (2023). Consistency models. arXiv preprint arXiv:2303.01469.

[3] Karras, T., Aittala, M., Aila, T., & Laine, S. (2022). Elucidating the design space of diffusion-based generative models. Advances in neural information processing systems, 35, 26565-26577.

**Questions:**

Please refer to the weakness section.

---

### Official Review · Reviewer_ZnHi · 2024-11-04

**Soundness:** 3
**Presentation:** 3
**Contribution:** 2
**Rating:** 5
**Confidence:** 5

**Summary:**

The paper introduces Multistep Consistency Models (MCMs) as an interpolation between Consistency Models (CMs) and Diffusion Models (DMs), dividing the time axis into multiple segments and applying a consistency tuning/distillation objective within each segment. Additionally, the Adjusted DDIM (aDDIM) is presented as an efficient deterministic sampling method that not only enhances synthetic sample quality but also improves MCM results when incorporated into the training algorithm.

**Strengths:**

The aDDIM sampler addresses the oversmoothing problem in deterministic sampling mechanisms, achieving performance comparable to that of a second-order sampler. The paper also provides comprehensive experiments on class-conditioned ImageNet and text-to-image generation.

**Weaknesses:**

See questions below.

**Questions:**

1. Are there any insights into the choice of hyperparameters in aDDIM, such as $\eta$?

2. The time axis is divided into equal segments. Are there alternative segmentation strategies that could improve performance?

3. The experiments are conducted using consistency tuning/distillation, where the model is initialized directly from a pretrained Diffusion Model, providing a good starting point. Could the proposed method also work with consistency training, i.e., random initialization?

---

### Note · Authors · 2024-11-19

I have read and agree with the venue's withdrawal policy on behalf of myself and my co-authors.